# Effects of SARS-CoV-2 Omicron BA.1 Spike Mutations on T-Cell Epitopes in Mice

**DOI:** 10.3390/v15030763

**Published:** 2023-03-16

**Authors:** Yudong Wang, Busen Wang, Zhenghao Zhao, Jinghan Xu, Zhe Zhang, Jinlong Zhang, Yi Chen, Xiaohong Song, Wanru Zheng, Lihua Hou, Shipo Wu, Wei Chen

**Affiliations:** Beijing Institute of Biotechnology, Beijing 100071, China

**Keywords:** SARS-CoV-2, Omicron BA.1, T-cell immunity, epitope

## Abstract

T-cell immunity plays an important role in the control of SARS-CoV-2 and has a great cross-protective effect on the variants. The Omicron BA.1 variant contains more than 30 mutations in the spike and severely evades humoral immunity. To understand how Omicron BA.1 spike mutations affect cellular immunity, the T-cell epitopes of SARS-CoV-2 wild-type and Omicron BA.1 spike in BALB/c (H-2^d^) and C57BL/6 mice (H-2^b^) were mapped through IFNγ ELISpot and intracellular cytokine staining assays. The epitopes were identified and verified in splenocytes from mice vaccinated with the adenovirus type 5 vector encoding the homologous spike, and the positive peptides involved in spike mutations were tested against wide-type and Omicron BA.1 vaccines. A total of eleven T-cell epitopes of wild-type and Omicron BA.1 spike were identified in BALB/c mice, and nine were identified in C57BL/6 mice, only two of which were CD4^+^ T-cell epitopes and most of which were CD8^+^ T-cell epitopes. The A67V and Del 69-70 mutations in Omicron BA.1 spike abolished one epitope in wild-type spike, and the T478K, E484A, Q493R, G496S and H655Y mutations resulted in three new epitopes in Omicron BA.1 spike, while the Y505H mutation did not affect the epitope. These data describe the difference of T-cell epitopes in SARS-CoV-2 wild-type and Omicron BA.1 spike in H-2^b^ and H-2^d^ mice, providing a better understanding of the effects of Omicron BA.1 spike mutations on cellular immunity.

## 1. Introduction

The SARS-CoV-2 Omicron variant was first detected in South Africa and Botswana in November 2021 and was listed by the WHO as a variants of concern (VOC) [1]. Omicron has since expanded to more than 100 subfamilies, such as BA.1, BA.2, BA.2.12.1, BA.4 and BA.5, and become globally dominant. Omicron subvariants BQ.1, BQ.1.1 and XBB are rapidly circulating in the United States, France, Singapore and India [2]. XBB.1.5 has become the dominant strain in the United States and is highly likely to cause the next wave of global coronavirus infections [3].

Spike is the most important functional protein for the entry and infection of the new coronavirus, and it is also the main antigen of many vaccines [4,5,6,7]. Omicron BA.1 contains more than 30 mutations in the spike protein, including G142D in the N-terminal domain (NTD); G339D, S371L/F, S373P, S375F, K417N, N404K, S477N, T478K, E484A, Q498R, N501Y and Y505H in the receptor-binding domain (RBD); and H655Y, N679K, P681H, N764K, D796Y, Q954H and N969K in the S2 subunit [8]. These mutations have resulted in Omicron’s significant humoral immune evasion in COVID-19 recovery patients and the vaccinated. The serum-neutralizing antibodies against Omicron in routinely vaccinated subjects are significantly lower than the wild-type (WT) virus and even below the lower limit of detection in some subjects [9,10,11,12,13]. Antibody escape against Omicron subvariants BQ and XBB was reported to be more significant, and serum antibodies against BQ and XBB were decreased by 13~81 times and 66~155 times, respectively [14].

Fortunately, unlike humoral immunity, T-cell immunity has a large cross-protective effect on Omicron. The T-cell immune response induced by SARS-CoV-2 infection or vaccination plays a key role in clearing the virus and generating T-cell memory. Specifically, CD4^+^ T-cell subsets support the production of anti-SARS-CoV-2 antibodies; cytotoxic CD8^+^ T cells also have a protective effect against infection [15], and T-cell responses are associated with rapid viral clearance and reduced disease severity. T cells recognize viral proteins extensively, with an estimated 30 epitopes spanning the entire SARS-CoV-2 proteome in each individual [16]. This broad recognition can limit the impact of individual viral mutations and may play an important role in protection against severe disease from viral variants [17]. Studies have shown that T-cell immunity induced by vaccination or previous infection is highly preserved among SARS-CoV-2 variants [18,19,20,21,22,23]. In infected patients or vaccination subjects, T-cell immunity to the Omicron strain was not significantly reduced, and 70~90% of CD4^+^ and CD8^+^ T cells responded to Omicron with no changed in the spike protein. T-cell epitope profiling showed that CD4^+^ and CD8^+^ T cells recognized a median of 11 and 10 epitopes in the spike protein, respectively, while the average preservation rate of Omicron was greater than 80% [20]. Although T-cell immunity is conservative against Omicron strains, 20% of subjects are >50% less responsive to Omicron spike, mainly in CD8^+^ T cells, which may be due to mutations that weaken binding to HLA I molecules [21], and 60% of CD4^+^ T-cell epitopes involved in Omicron mutations have been shown to be manifest as complete or partial loss of T-cell response [24]. Human HLA typing is diverse, so the T-cell epitopes recognized by each person also differ [16]; therefore, it is difficult to interpret the effect of mutations on T-cell immunity from specific T-cell epitopes.

Mice are important model animals, with relatively few MHC molecules, facilitating the study of T-cell epitopes. Here, we analyzed the cellular immunity differences in BALB/c and C57BL/6 mice vaccinated with an adenovirus type 5 vector encoding the SARS-CoV-2 WT spike (Ad5-Spike-WT) or the Omicron BA.1 spike (Ad5-Spike-BA.1), identified the T-cell epitopes of the WT and Omicron BA.1 spike and demonstrated that the mutations in spike resulted in several changes in the epitopes, while most of the epitopes were conserved.

## 2. Materials and Methods

### 2.1. Vaccine and Mice

The SARS-CoV-2 vaccines Ad5-Spike-WT (also known as Ad5-nCoV) and Ad5-Spike-BA.1 were prepared by CanSino Biotechnology Inc. (Tianjin, China), as described in previous studies [7,25]. Specific-pathogen-free wild-type BALB/c (H-2^d^) and C57BL/6 (H-2^b^) mice were obtained from Beijing Vital River Laboratory Animal Technologies Co., Ltd. (Beijing, China) and SPF Biotechnology Co., Ltd. (Beijing, China), respectively. The mice were housed and bred in the temperature-, humidity- and light-cycle-controlled animal facility (20 ± 2 °C; 50 ± 10%; light, 7:00–19:00; dark, 19:00–7:00) of the Animal Center, Academy of Military Medical Sciences, Beijing, China. BALB/c and C57BL/6 mice between 6 and 8 weeks of age were intramuscularly injected with 5 × 10^8^ viral particles (VP) of Ad5-Spike-WT or Ad5-Spike-BA.1 in a volume of 100 µL. The experiments involving animals were approved by the Institutional Experimental Animal Welfare and Ethics Committee and were conducted in accordance with its guidelines.

### 2.2. Splenocyte Isolation

The isolation of splenocytes was performed as described previously [26]. Briefly, mice were euthanized on day 14 post immunization, and spleens were pushed through a 70 μm cell strainer in complete RPMI1640 medium to prepare a single-cell suspension under aseptic conditions. The cells were centrifuged at 500× *g* for 5 min, resuspended in 3 mL RBC lysis solution (Sigma, Rahway, NJ, USA) and incubated for 5 min to lyse the red blood cells. After two cycles of washing in complete RPMI 1640 medium, the splenocytes were counted and kept on ice until required.

### 2.3. Peptide Synthesis and Peptide Pool Preparation

Peptides (15 amino acids in length, overlapping by 11 amino acids) encompassing SARS-CoV-2 wild-type and Omicron BA.1 spike, as well as all truncated 9-mer peptides, were synthesized by GL Biochem Ltd. (Shanghai, China). The 15-mer peptides were based on the amino acid sequence of Wuhan-Hu-1 (GenBank identifier: NC_045512.2) and Omicron BA.1 (GISAID: EPI_ISL_6640917, Figure 1A), which were identical to the spike sequences of the corresponding vaccines used in this study. The truncated 9-mer MHC-I-restricted epitopes were derived from the positive CD8^+^ T-cell 15-mer peptides and predicted for MHC-I binding by “IEDB recommended 2020.02 (NetMHCpan EL4.1)” in the Immune Epitope Database (http://tools.iedb.org/mhci/, accessed on 8 August 2022). MHC alleles of H2-K^d^, H2-D^d^ and H2-L^d^ for BALB/c and H2-K^b^ and H2-D^b^ for C57BL/6 were predicted, and 3~5 peptides with the highest scores in each positive 15-mer peptide were selected. All peptides were provided at >90% purity, as verified by high-performance liquid chromatography. The two-dimensional peptide pools were divided into 4 groups (NTD-1~18, RBD-1~16, S1-1~12 and S2-1~24), with a total of 70 pools (Appendix A). The NTD-1~18 pools contained peptides 1~73, the RBD-1~16 pools contained peptides 74–132, the S1-1~12 pools contained peptides 133~168 and the S2-1~24 pools contained the remaining peptides. Each peptide appeared in two pools. Peptides were dissolved in dimethyl sulfoxide (DMSO) at approximately 12 mg/mL, and pools of peptides containing 10~12 peptides had at a final concentration of 1 mg/mL. The peptides were frozen at −20 °C until required.

### 2.4. Screening of T-Cell Epitopes by ELISpot

Two-dimensional matrix pools were designed for screening of the T-cell epitopes. The overlapping peptides of the WT and Omicron BA.1 spike were screened and further verified with individual peptides. The 316 overlapping peptides were coded, mixed in 70 matrix peptide pools (Appendix A) and detected using an IFNγ ELISpot kit (Mabtech, Nacka Strand, Sweden). ELISpot plates were coated overnight at 4 °C with 5 μg/mL of anti-mouse IFNγ antibody. The antibody-coated plates were washed five times with sterile PBS and blocked with complete RPMI1640 medium for 2 h at room temperature. After blocking, 100 μL of splenocyte suspension (2 × 10^6^ cells/mL) containing matrix peptide pools (1 μg/mL) or individual peptide (10 μg/mL) were added to each well. A ‘no peptide’ negative control was included in all assays. The plates were incubated for 18–24 h at 37 °C/5% CO_2_. Following incubation, the wells were washed five times with PBS. Biotinylated anti-mouse IFNγ was added to each well at a concentration of 1 μg/mL and incubated for 2 h at room temperature. Following three washes, streptavidin–horseradish peroxidase was added to each well and incubated for 1 h. After five washes with PBS, the colorimetric reactions were developed using 3,3′,5,5′-tetramethylbenzidine as a substrate. Upon visualization of the spots, the reaction was stopped by rinsing in tap water. Membranes were allowed to dry overnight in the dark; then, spots were counted with an AT-Spot 3200 (SinSage Technology, Beijing, China). Results were expressed as the number of spot-forming cells (SFCs) per 10^6^ splenocytes and considered positive if the magnitude of the response was SFCs > 50 and the magnitude of the positive was 2-fold greater than the control well.

### 2.5. Intracellular Cytokine Staining

An intracellular cytokine staining (ICS) assay was conducted as previously described [27]. Briefly, the splenocytes were stimulated for 6 h at 37 °C with 1 μg/mL of peptide pools, 10 μg/mL of a selected single peptide or the same volume of DMSO as the background control and with BD GolgiStop^TM^ to block cytokine secretion. A positive control with ionomycin and PMA was included in each assay. Following stimulation, the cells were washed and stained with Near-IR viability dye (Thermo Fisher Science, Waltham, MA, USA) for 20 min to exclude dead cells from data analysis. After one wash with PBS, the splenocytes were incubated with a mixture of antibodies against lineage markers, including anti-CD3 PerCP-Cy5.5 (clone 17A2), anti-CD4 Alexa Fluor 700 (clone RM4-5) and anti-CD8 FITC (clone 5H10-1). After one wash with PBS, the cells were fixed and permeabilized with Cytofix/Cytoperm (BD Biosciences, San Diego, CA, USA), washed with Perm/Wash buffer (BD Biosciences, San Diego, CA, USA) and incubated with anti-IFNγ PE (clone XMG1.2). The cells were washed successively with Perm/Wash buffer and PBS, an data were acquired on a FACS Canto^TM^ (BD Biosciences, San Diego, CA, USA).

### 2.6. Statistical Analysis

Data are expressed as the mean ± SEM. For all analyses, *p*-values were analyzed with paired *t* test (n.s. *p* > 0.05; * *p* < 0.05; ** *p* < 0.01; *** *p* < 0.001; ****, *p* < 0.0001). All graphs were analyzed with Prism software version 9.0 (GraphPad Software, Inc. San Diego, CA, USA).

## 3. Results

### 3.1. Ad5-Spike-WT and Ad5-Spike-BA.1 Induced Robust but Different Cellular Immune Responses in Mice

More than 30 amino acid mutations were identified within the spike of the Omicron BA.1 variant (Figure 1A). In order to test the immune response caused by the mutations, BALB/c and C57BL/6 mice were vaccinated with Ad5-Spike-WT or Ad5-Spike-BA.1, and the cellular immune response of the vaccine candidates against both WT and Omicron BA.1 spike were detected by IFNγ ELISpot and ICS assays. Both the Ad5-Spike-WT and Ad5-Spike-BA.1 induced robust IFNγ responses specific to WT and Omicron BA.1 spike (Figure 1B–G). In BALB/c mice, Ad5-Spike-WT induced more IFNγ spots specific to WT spike, and Ad5-Spike-BA.1 induced more IFNγ spots specific to Omicron BA.1 spike (Figure 1B). These differences were reflected in a higher CD4^+^ IFNγ response specific to WT spike in mice vaccinated with Ad5-Spike-WT and a higher CD8^+^ IFNγ response specific to Omicron BA.1 spike in mice vaccinated with Ad5-Spike-BA.1 (Figure 1C,D). However, in C57BL/6 mice, both Ad5-Spike-WT and Ad5-Spike-BA.1 induced more IFNγ spots specific to Omicron BA.1 spike (Figure 1E). Furthermore, a higher CD8^+^ IFNγ response specific to Omicron BA.1 spike (Figure 1F) and a higher CD4^+^ IFNγ response specific to WT spike (Figure 1G) were induced by Ad5-Spike-WT. The difference in the cellular immune response induced by Ad5-Spike-WT and Ad5-Spike-BA.1 indicates that the mutations of the spike caused changes in T-cell epitopes.

### 3.2. Identification of T-Cell Epitopes in WT and Omicron BA.1 Spike

In order to identify the T-cell epitopes altered by Omicron BA.1 spike mutations, the epitopes restricted by H-2D/K/L^d^ (MHC class I haplotype for BALB/c), H-2K/D^b^ (MHC class I haplotype for C57BL/6), I-A/E^d^ (MHC class II haplotype for BALB/c) and I-A^b^ (MHC class II haplotype for C57BL/6) in WT and Omicron BA.1 spike were identified. The peptides covering the complete mature sequences of Spike were synthesized, with 316 peptides for WT and 315 for Omicron BA.1, all of which were 15-mer sequences with an overlap of 11 amino acids. They were divided into four subgroups (NTD-1~18, RBD-1~16, S1-1~12 and S2-1~24) to construct two-dimensional peptide pools, with each peptide appearing in two pools (Appendix A). The responsive peptide pools were screened out by an IFNγ ELISpot assay, with the peptides on the cross position of the two positive pools identified as probable epitopes and verified in a further assay. In BALB/c mice inoculated with Ad5-Spike-WT, peptide pools of NTD-2, NTD-8, NTD-14, NTD-16, RBD-2, RBD-7, RBD-8, RBD-10, RBD-11, RBD-13, RBD-14, RBD-16, S1-1, S1-7, S1-8, S2-8 and S2-23 produced a positive reaction. Based on the peptide–matrix setup analysis, 19 individual probable epitope candidates were identified (Appendix A). A further validation identified the peptides of W61-75, W269-283, W353-367, W501-515, W505-519, W521-535, W525-539, W529-543, W533-547 and W1049-1063 as positive (Figure 2A); only W1049-1063 was in the S2 subunit, and all the other epitopes were in the S1 regions. The frequencies of peptide-specific IFNγ-producing T cells ranged from 68 to 498 SFCs per 10^6^ splenocytes.

For Ad5-Spike-BA.1, peptides pools of NTD-9, NTD-11, NTD-12, RBD-6, RBD-7, RBD-8, RBD-9, RBD-10, RBD-13, RBD-14, S1-1, S1-7, S2-8, S2-22 and S2-23 produced a positive reaction (Appendix A), and the peptides of O261-275, O265-279, O469-483, O473-487, O485-499, O501-515, O521-535, O529-543, O1045-1059 and O1049-1063 were positive (Figure 2B). Similar to the epitopes of WT spike, most of the positive peptides were in the S1 subunit, and the frequencies of peptide-specific IFNγ-producing T cells ranged from 108 to 535 SFCs per 10^6^ splenocytes.

With the same strategy, the H-2K/D^b^- or I-A^b^-restricted epitopes were identified in C57BL/6 mice (Appendix A); W61-75, W257-271, W261-275, W389-403, W505-519, W509-523, W533-547, W537-551 and W741-755 were the responsive peptides in WT spike (Figure 2C), and O253-267, O257-271, O501-515, O505-519, O529-543, O533-547, O641-655, O1169-1183 and O1173-1187 were the responsive peptides in Omicron BA.1 spike (Figure 2D). Similar to BALB/c mice, most of the T-cell epitopes in C57BL/6 mice were in the S1 subunit. The frequencies of peptide-specific IFNγ-producing T cells ranged from 201 to 1745 SFCs per 10^6^ splenocytes, which is a higher range than that in BALB/c mice.

### 3.3. Characterization of the CD8^+^ and CD4^+^ T-Cell Epitopes

To further determine which T-cell subsets were activated by these peptides, the splenocytes were stimulated with a single peptide, and the IFNγ secretion the antigen-specific CD8^+^ and CD4^+^ T cells was examined using an ICS assay. For BALB/c mice, W61-75 and W353-367 stimulated CD4^+^ T cells, while W269-283, W501-515, W505-519, W521-535, W525-539, W529-543, W533-547 and W1049-1063 stimulated CD8^+^ T cells (Figure 3A and Appendix A). However, all the responsive peptides in Omicron BA.1 spike were recognized as CD8^+^ T cell epitopes (Figure 3B and Appendix A). For C57BL/6 mice, the CD4^+^ T-cell-specific peptide in WT spike was W61-75; the CD8^+^ T-cell-specific peptides in WT spike were W257-271, W261-275, W389-403, W505-519, W509-523, W533-547, W537-551 and W741-755 (Figure 3C and Appendix A); and all the responsive peptides in Omicron BA.1 spike were CD8^+^ T-cell-specific (Figure 3D and Appendix A). Taken together, these results demonstrate that both WT and Omicron BA.1 spike contain more CD8^+^ T-cell epitopes in splenocytes of BALB/c and C57BL/6 mice.

To further identify the exact short epitopes recognized by CD8^+^ T cells within the overlapping 15-mer peptides, we predicted the potential 9-mer epitopes for MHC-I binding in the Immune Epitope Database (http://tools.iedb.org/mhci/, accessed on 8 August 2022) as determined by IFNγ-ELISpot (Figure 4, Appendix A). A total of three H-2D^d^-, two H-2K^d^-, four H-2L^d^-, four H-2D^b^- and four H-2K^b^-restricted epitopes were identified (Table 1). Two I-E^d^-restricted epitopes and one I-A^b^-restricted epitope with 15 amino acids were also identified (Table 1). For the T-cell epitopes of WT spike, W61-75 and W505-513 involved Omicron BA.1 mutations, and the rest of the epitopes were unaffected in BALB/c mice, while only W61-75 was involved in Omicron BA.1 mutations in C57BL/6 mice. This suggests that most of the T-cell epitopes in BALB/c and C57BL/6 mice are highly preserved among WT and Omicron BA.1. We also identified some T-cell epitopes only present in Omicron BA.1 spike, including O475-483, O486-494 and O502-510 in BALB/c mice and O645-653 in C57BL/6 mice, which were derived from Omicron BA.1 spike mutations.

### 3.4. The Effect of Spike Mutations on T-Cell Epitopes

T-cell epitopes of WT spike were compared with those identified in Omicron BA.1 spike in BALB/c and C57BL/6 mice. The epitopes involving Omicron BA.1 spike mutations are summarized in Figure 5A. Those peptides were further verified in the mice vaccinated with Ad5-Spike-WT and Ad5-Spike-BA.1 to determine the effect of amino acid mutation on the epitopes. In Ad5-Spike-WT-immunized BALB/c and C57BL/6 mice, W61-75 was the core sequence of dominant CD4^+^ T-cell epitopes of WT spike but was unresponsive in Ad5-Spike-BA.1-immunized mice, whereas O61-73, the corresponding peptide of W61-75 in Omicron BA.1 spike, was not responsive in either Ad5-Spike-WT- or Ad5-Spike-BA.1-immunized mice (Figure 5B,C), implying that the A67V and Del 69-70 mutation abrogated this T-cell epitope. On the other hand, Omicron BA.1 spike mutations formed new epitopes that were not present in WT spike. The T478K and E484A mutations in peptide W478-486 resulted in a new H-2L^d^ epitope, O475-483, and the Q493R and G496S mutations in W489-497 resulted in a new H-2L^d^ epitope, O486-494, in BALB/c mice (Figure 4B). Similarly, the H655Y mutation in W648-656 resulted in a new peptide, O645-653, which induced T-cell response in C57BL/6 mice vaccinated with Ad5-Spike-BA.1 (Figure 5C). Furthermore, some mutations had no effect on the T-cell responses, for example, there was no difference in the induction of T-cell responses between W505-513 and O502-510, indicating that Y505H mutation did not affect the T-cell epitope (Figure 5B).

## 4. Discussion

In this study, we found that the cellular immune responses of Ad5-Spike-BA.1 in BALB/c and C57BL/6 mice were robust but differed from those of Ad5-Spike-WT, and the T-cell epitopes of the spike in WT and Omicron BA.1 variant were identified.

Nine MHC-I-restricted epitopes and two MHC-II-restricted epitopes in BALB/c mice, as well as eight MHC-I-restricted epitopes and one MHC-II-restricted epitope in C57BL/6 mice, were identified in the current study. MHC-I-restricted epitopes account for the majority, which is consistent with the T-cell response of the Ad5-vectored SARS-CoV-2 vaccine in mice, i.e., that CD8^+^ T cells dominate the spike-specific response [27]. This phenomenon was also reported in the ChAdOx1-vectored SARS-CoV-2 vaccine and mRNA vaccines [28,29,30]. However, those vaccines elicited a robust CD4^+^ T-cell response in humans [7,31,32]. The difference in T-cell response between mice and humans may be due to differences in the immune system and the diversity of MHC molecules between species.

There are more than 30 mutations on the Omicron BA.1 spike, almost half of which are located in the RBD. These mutations in the spike largely attenuated the protective ability of humoral immunity developed against previous strains. However, several studies suggested that in donors who were vaccinated with wild-type spike or infected by previous strains, the T-cell responses against the Omicron variant were largely maintained. At the epitope level, T-cell epitopes of wild-type SARS-CoV-2 are considerably preserved across major Omicron subvariants [33,34,35]. The currently circulating XBB.1.5, BF.7 and BQ.1 variants exhibit stronger humoral immune evasion than BA.1, but it does not have more mutations against the T-cell epitope in mice.

Antigen recognition by T cells involves the interaction of three classes of molecules: MHC, T-cell epitopes (antigenic peptides) and T-cell receptors. T-cell epitopes generally have two or more sites that bind to specific MHC molecules, and these amino acids bind to MHC molecules via hydrogen bonds and are then recognized by T-cell receptors. BALB/c and C57BL/6 mice possess H-2K/D/L^d^ and H-2K/D^b^ MHC class I molecules, respectively, and no identical MHC-I-restricted epitope of the SARS-CoV-2 Spike was found between the two mice; however, an identical MHC-II-restricted epitope of SARS-CoV-2 spike was identified. This I-A/E^d^- and I-A^b^-restricted epitope, W61-75, has also been described to show T-cell responsiveness in humans [24]. Peptides that bind to MHC class II molecules contain an internal sequence of 7 to 10 amino acids that provides the major contact points, and the class II pocket shows fewer restricted amino acid sequence preferences. Furthermore, the hydrogen bonds between the backbone of the peptide and the class II molecule are distributed throughout the binding site rather than being clustered predominantly at the ends of the site. It is likely that the internal 7~10 amino acids of W61-75 can provide major contact points that can interact well with MHC-II molecules of both mice and humans.

Mutations at different locations on the epitope have different effects on the function of the T-cell epitopes. For example, the Y505H mutation, which is localized on the first amino acid position of the peptide, does not affect the T-cell response. However, T478K, E484A, Q493R and G496S mutations form two new epitopes in BALB/c mice, and the H655Y mutation forms one new epitope in C57BL/6 mice. It is likely that those mutations contribute important anchor residues to the epitopes. Some mutations cause T-cell epitope inactivation, A67V and Del 69-70 resulted in the incapacitation of W61-75 in both BALB/c and C57BL/6 mice in this study, as well as in humans [24]. Similarly, one P272L mutation of wild-type CD8^+^ T-cell epitopes, Spike269-277 YLQPRTFLL, allows the virus to evade T-cell responses in the HLA A*02 convalescent patients and individuals vaccinated against SARS-CoV-2 [36].

Some CD8^+^ T-cell epitopes identified in our work have been previously reported in mice infected with SARS-CoV-2 or vaccinated with the SARS-CoV-2 Spike RBD subunit vaccine [37,38]. The conserved Spike539-546 VNFNFNGL and Spike511-518 VVLSFELL are common CD8^+^ T-cell epitopes within SARS-CoV-2 and SARS-CoV and were also found in the C57BL/6 mice and BALB/c mice vaccinated with Ad5-Spike-WT and Ad5-Spike-BA.1 in our work. Spike539-546 VNFNFNGL was validated to be fully protective in K18-ACE2 mice infected with SARS-CoV-2 [39]. This suggests that these CD8^+^ T-cell epitopes identified by viral infection or vaccination with different types of vaccine were consistent and that these conserved T-cell epitopes play an important role in cross protection against SARS-CoV-like strains.

Previous infection usually negatively impacts the response to a subsequent infection with pathogens sharing cross-reactive antigens. The initial immune background places restrictions on the antibody response induced by a subsequent SARS-CoV-2 variant [40,41,42], but it seems to show limited impact on recalling CD8^+^ T-cell responses [43]. The spike mutation of Omicron BA.1 induced two new MHC-I-restricted epitopes in BALB/c mice and one new MHC-I-restricted epitope in C57BL/6 mice. It is likely that a bivalent SARS-CoV-2 vaccine containing Omicron spike may induce a broader T-cell response. However, the T-cell response against the new Omicron epitopes after previous exposure or vaccination with WT spike was not tested in this study, and the effect of Omicron spike mutations on T-cell epitopes in humans needs to be studied.

In summary, we identified the T-cell epitopes of SARS-CoV-2 WT and Omicron BA.1 spike in BALB/c and C57BL/6 mice, and the results serve contribute to our understanding of the effects of Omicron BA.1 spike mutations on T-cell immunity.

## Figures and Tables

**Figure 1 viruses-15-00763-f001:**
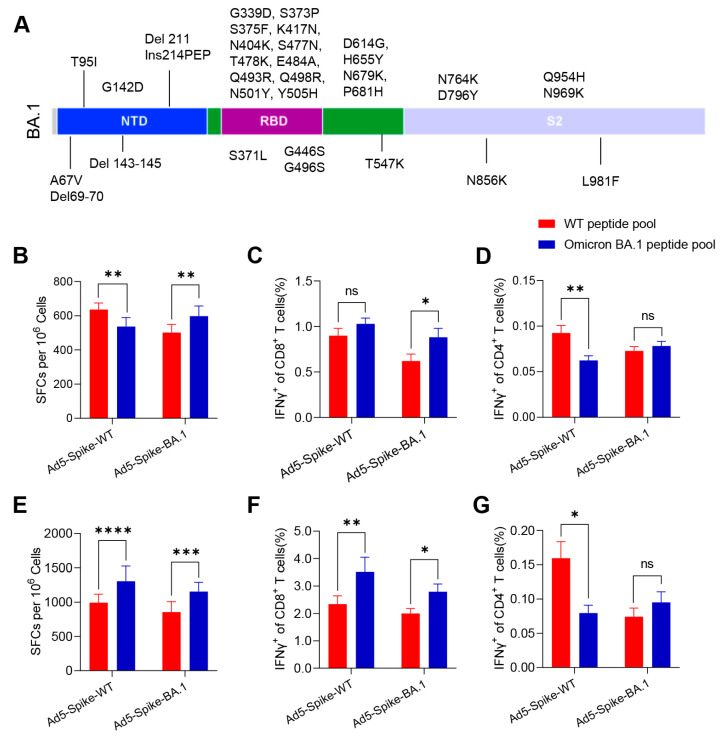
The specific cellular immune responses in BALB/c and C57BL/6 mice vaccinated with Ad5-Spike-WT and Ad5-Spike-BA.1. (**A**). The amino acid mutations of Omicron BA.1 spike compared to the WT. (**B**–**G**) The cellular immune responses induced by Ad5-Spike-WT, Ad5-Spike-BA.1. BALB/c (**B**–**D**) (*n* = 10 per group) and C57BL/6 (**E**–**G**) mice (*n* = 6 per group) intramuscularly injected with 5 × 10^8^ VP of Ad5-Spike-WT or Ad5-Spike-BA.1; the splenocytes were prepared 2 weeks after vaccination and stimulated with WT spike peptide pool (red) or Omicron BA.1 spike peptide pool (blue), and the cellular immune responses were detected by IFNγ ELISpot assay (**B**,**E**), CD8^+^ IFNγ intracellular staining (**C**,**F**) and CD4^+^ IFNγ intracellular staining (**D**,**G**). Data are presented as mean ± SEM. Statistical significance was determined by paired *t* tests. ns, *p* > 0.05; *, *p* < 0.05; **, *p* < 0.01; ***, *p* < 0.001; ****, *p* < 0.0001. SFCs = spot-forming cells.

**Figure 2 viruses-15-00763-f002:**
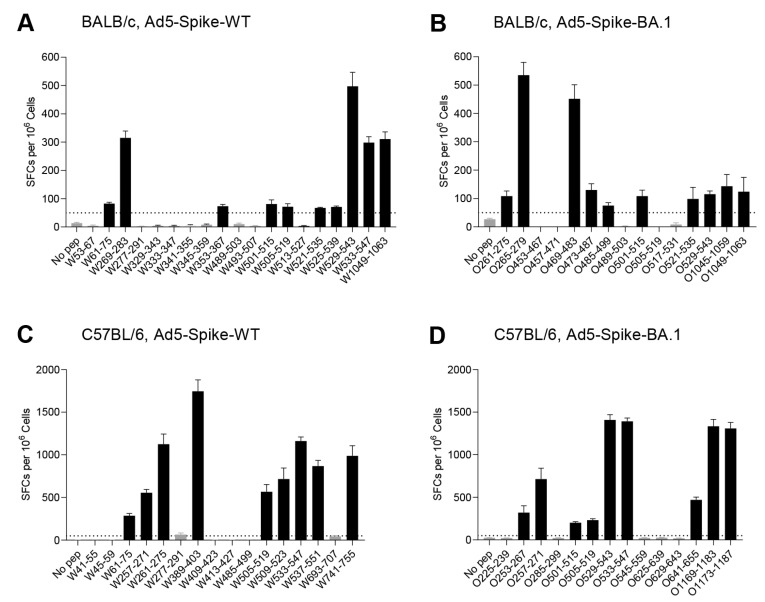
Identification of T-cell epitopes of WT and Omicron BA.1 spike in BALB/c and C57BL/6 mice. The second-round screening of T-cell epitopes was performed according to the peptide matrix of the first-round screening (Appendix A). BALB/c and C57BL/6 mice (*n* = 6 per group) were inoculated with 5 × 10^8^ VP of Ad5-Spike-WT (**A**,**C**) or Ad5-Spike-BA.1 (**B**,**D**). The splenocytes were collected 2 weeks after vaccination for an IFNγ ELISpot assay. (**A**) A total of 19 peptides from WT spike and (**B**) 15 peptides from Omicron BA.1 spike were identified in BALB/c mice, while (**C**) 16 peptides from WT spike and (**D**) 14 peptides from Omicron BA.1 spike were identified in C57BL/6 mice. Each bar represents the mean ± SEM. Thresholds (SFCs = 50 per 10^6^ cells) for positivity are indicated by black dashed lines in all panels.

**Figure 3 viruses-15-00763-f003:**
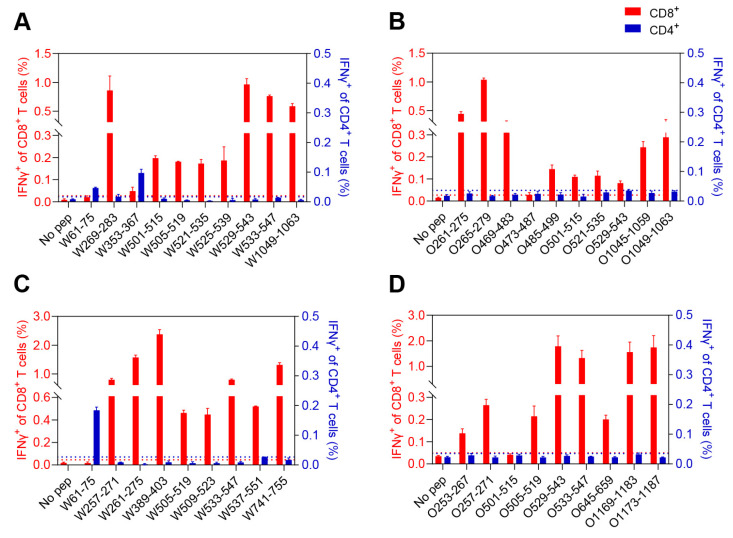
Characterization of the T-cell epitopes by ICS assay in BALB/c and C57BL/6 mice. BALB/c and C57BL/6 mice (*n* = 3 per group) were inoculated with 5 × 10^8^ VP of Ad5-Spike-WT or Ad5-Spike-BA.1. The splenocytes were harvested 14 days post immunization and stimulated with specific peptide to assess cytokine production of IFNγ by ICS. The percentages of cytokine-secreting cells were calculated, and the results were summarized in (**A**) 10 peptides of WT spike in BALB/c mice, (**B**) 10 peptides of Omicron BA.1 spike in BALB/c mice, (**C**) 9 peptides of WT spike in C57BL/6 mice and (**D**) 9 peptides of Omicron BA.1 spike in C57BL/6 mice. Thresholds for positivity were mean + 3 SD of all values from mock-vaccinated mice. Red dashed line represents the positive threshold of IFNγ response in CD8^+^ T cells; blue dashed line represents the positive threshold of IFNγ response in CD4^+^ T cells. Each bar represents the mean ± SEM of three replicates.

**Figure 4 viruses-15-00763-f004:**
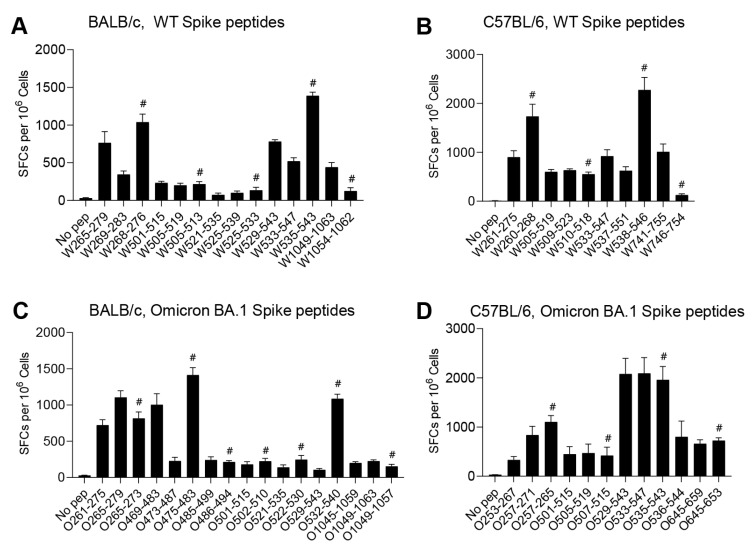
Identification of the exact CD8^+^ T-cell epitopes in BALB/c and C57BL/6 mice. BALB/c and C57BL/6 mice (*n* = 3 per group) were vaccinated with Ad5-Spike-WT or Ad5-Spike-BA.1; the splenocytes were collected 2 weeks after vaccination and stimulated with positive peptides and corresponding truncated 9-mer peptides to assess T-cell responses by IFNγ ELISpot. (**A**,**C**) The splenocytes from BALB/c mice vaccinated with Ad5-Spike-WT (**A**) and Ad5-Spike-BA.1 (**C**) were stimulated with the 15-mer peptides and the corresponding 9-mer peptides. (**B**,**D**) The splenocytes from C57BL/6 mice vaccinated with Ad5-Spike-WT (**B**) and Ad5-Spike-BA.1 (**D**) were stimulated with the 15-mer peptides and the corresponding 9-mer peptides. T-cell responses were determined by IFNγ ELISpot assays. The truncated epitopes are labeled with #. All results are expressed as mean ± SEM.

**Figure 5 viruses-15-00763-f005:**
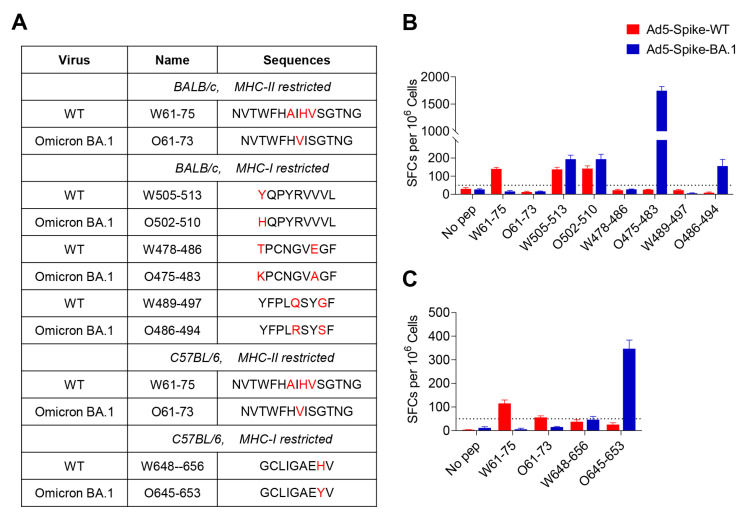
The effect of spike mutations on T-cell epitopes. (**A**) T-cell epitopes involving Omicron BA.1 spike mutations in BALB/c and C57BL/6 mice. The red letters indicate the mutated amimo acids. (**B**,**C**) The cellular immune response of those epitopes was determined by IFNγ ELISpot assays in BALB/c (**B**) and C57BL/6 (**C**) mice (*n* = 6 per group). All results are expressed as mean ± SEM. Red, mice vaccinated with Ad5-Spike-WT; blue, mice vaccinated with Ad5-Spike-BA.1. Thresholds (SFCs = 50 per 10^6^ cells) for positivity are indicated by black dashed lines in both panels.

**Table 1 viruses-15-00763-t001:** Characteristics of SARS-CoV-2 spike-specific T-cell epitopes in BALB/c and C57BL/6 mice.

Name	Epitope	Protein	Start Position	End Position	CD4/CD8	MHC
**BALB/c**						
W61-75	NVTWFHAIHVSGTNG	WT	61	75	CD4	I-E^d^
W353-367	WNRKRISNCVADYSV	WT/Omicron BA.1	353	367	CD4	I-E^d^
W268-276	GYLQPRTFL	WT/Omicron BA.1	268	276	CD8	H2-L^d^
W505-513	YQPYRVVVL	WT	505	513	CD8	H2-K^d^
W525-533	CGPKKSTNL	WT/Omicron BA.1	525	533	CD8	H2-D^d^
W535-543	KNKCVNFNF	WT/Omicron BA.1	535	543	CD8	H2-L^d^
W1054-1062	QSAPHGVVF	WT/Omicron BA.1	1054	1062	CD8	H2-D^d^
O475-483	KPCNGVAGF	Omicron BA.1	475	483	CD8	H2-L^d^
O486-494	YFPLRSYSF	Omicron BA.1	486	494	CD8	H2-L^d^
O502-510	HQPYRVVVL	Omicron BA.1	502	510	CD8	H2-D^d^
O1049-1057	FPQSAPHGV	WT/Omicron BA.1	1049	1057	CD8	H2-K^d^
**C57BL/6**						
W61-75	NVTWFHAIHVSGTNG	WT	61	75	CD4	I-A^b^
W260-268	AGAAAYYVG	WT/Omicron BA.1	260	268	CD8	H2-K^b^
W389-403	DLCFTNVYADSFVIR	WT/Omicron BA.1	389	403	CD8	H-2-D^b^
W510-518	VVVLSFELL	WT/Omicron BA.1	510	518	CD8	H2-K^b^
W538-546	CVNFNFNGL	WT/Omicron BA.1	538	546	CD8	H2-K^b^
W746-754	STECSNLLL	WT/Omicron BA.1	746	754	CD8	H2-D^b^
O506-514	RVVVLSFEL	WT/Omicron BA.1	506	514	CD8	H2-K^b^
O645-653	GCLIGAEYV	Omicron BA.1	645	653	CD8	H2-D^b^
O1173-1187	VVNIQKEIDRLNEVA	WT/Omicron BA.1	1173	1187	CD8	H2-D^b^

## Data Availability

All data required to interpret the data are provided in the main document or the Appendix A. Further data are available from the corresponding author upon reasonable request.

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
