# Peer review of "Effects of SARS-CoV-2 Omicron BA.1 Spike Mutations on T-Cell Epitopes in Mice"

_viruses, 2023, doi:10.3390/v15030763_

Round 1

Reviewer 1 Report

Dear Editor

I would like to thank the authors for this important work.

In abstract , it is not clear if  a combination of wild SARS-CoV-2 and omicron vaccines was tested . Also, it is not clear if wide type vaccine was tested against Omicron and vice versa. Materials and Methods is perfect except for synthetic peptides, it  is not mentioned how it was predicted and using which tools. The results need to be more clear.

Regards

Amal Mahmoud

Author Response

Dear Editors and reviewers:

Thank you for your precious comments and advice. Those comments are all valuable and very helpful for revising and improving our paper, as well as the important guiding significance to our researches. We have studied comments carefully and have made correction which we hope meet with approval.

According to the editor and reviewer’s suggestions, we have now amended the relevant part in manuscript. Please see the attachment for a detailed response.

Thank you once again for your attention to our paper.

Best Regards.

Yours sincerely,

Shipo Wu

Reviewer 2 Report

Please find enclosed the attachment with reviewer's comments

Author Response

(The authors gave the same response as above.)

Reviewer 3 Report

This is a solid manuscript based on well-designed experiments and credible data, as well as it is carefully written and clearly presented. The results, while some of them have been previously reported, contribute to our understanding of the adenovirus vaccines of SARS-CoV-2 and the effects of Spike mutations on T cell immunity.

There are some minor ambiguities I would like to discuss:

1. The first sentence of abstract should be refined or rephrased.

2. line 41: It is unclear whether the word “respectively” decorates BQ/XBB or it decorates vaccinated/infected patients.

3. line 53: “highly conserved in the SARS-CoV-2 Spike” usually means lack of cross-variant protection, it is better to change to “highly preserved among SARS-CoV-2 variants”.

4. line 183: The first paragraph of section 3.2 is a very clever way to reduce the number of tests, but it should be treated carefully to avoid mistakes/typos. According to figure S2, there are 19 candidates, but one is missing and it only shows 18 in the manuscript as well as in Figure 2A. But this is an honest mistake and no big deal.

5. T cell epitopes varies from BALB/c mice to C57BL/6, let alone the tremendous differences between human and mice. This is the main shortness of this work, which obviously degrade the importance of this manuscript. Still, it is of benefit to discuss this issue in the discussion section. Line 314-316 hits the point, but more examples and deeper discussion should be added.

Author Response

(The authors gave the same response as above.)

Round 2

Reviewer 1 Report

Thanks for your Responses